# Selected Milestones in Antiviral Drug Development

**DOI:** 10.3390/v16020169

**Published:** 2024-01-23

**Authors:** Erik De Clercq

**Affiliations:** Rega Institute for Medical Research, KU Leuven, Herestraat 49, B-3000 Leuven, Belgium; erik.declercq@kuleuven.be

**Keywords:** HIV, HBV, HSV, VSV, CMV, AIDS, ANPs, NRTIs, NtRTIs, NNRTIs

## Abstract

This review article will describe the (wide) variety of approaches that I envisaged to develop a specific therapy for viral infections: (i) interferon and its inducers, (ii) HSV, VZV and CMV inhibitors, (iii) NRTIs (nucleoside reverse transcriptase inhibitors), NtRTIs (nucleotide reverse transcriptase inhibitors) and NNRTIs (non-nucleoside reverse transcriptase inhibitors) as HIV inhibitors, (iv) NtRTIs as HBV inhibitors, and finally, (v) the transition of an HIV inhibitor to a stem cell mobilizer, as exemplified by AMD-3100 (Mozobil^®^).

## 1. Introduction

In starting a scientific career with Prof. Piet De Somer at the Rega Institute, I initially dreamed of discovering a cure for cancer. This ambitious goal was fueled by two expectations. The first one was the hope that interferon, which had been recognized as an antiviral substance in 1957 by Isaacs and Lindenmann [1], generated sudden attention in the 1970s as a potential anticancer agent under the impulse of Dr. Mathilde Krim (International Workshop on Interferon in the Treatment of Cancer, Memorial Sloan-Kettering Cancer Center, New York, USA, 31 March–2 April 1975). A second stimulus emanated from the discovery of the RT [reverse transcriptase (RNA-dependent DNA polymerase)] by Temin and Mizutani [2], and Baltimore [3], subsequently confirmed in several other laboratories (including ours) all over the world. The work of Sol Spiegelman and his co-workers who detected RT activity in virtually all human cancers that they looked at [4,5,6,7,8,9,10,11,12,13,14,15] caused such a hype that I also started to look for RT inhibitors as potential anticancer agents. As a result, I found suramin to be a potent inhibitor of RT in murine tumor viruses [16].

## 2. Interferon Inducers (Figure 1)

Polyacrylic acid (PAA) and polymethacrylic acid (PMEA) were discovered to be interferon inducers [17,18] shortly after pyran copolymer had been reported as an inducer of IFN by Merigan [19]. Merigan then reported that pyran copolymer acted as an inducer of IFN in mice and men [19,20]. That polyanionic substances were able to act as inducers of endogenous IFN was further substantiated by the interferon-inducing capacity of PAA [18] in vivo. IFN, through the induction by PAA, also protected newborn mice against the lethal effects of a rhabdo (VSV) infection [21] (which later on was also shown to serve as a surrogate virus for Ebolavirus infection [22]). The endogenous IFN induced by PAA as well as exogenous IFN were shown to inhibit the formation of, i.e., vaccinia virus-induced tail lesions in mice [23], a paradigm for the inhibitory effects of PAA and interferon on poxvirus infections [24]. The prolonged antiviral protection offered by PAA [25] should have stimulated a further evaluation of the antiviral propensity of PAA when administered in the course of various other virus infections, but the long-term toxicity potential of PAA may have prohibited this further evaluation. Instead, the polyacetal carbocyclic acids, i.e., COAM (chlorite-oxidized oxyamylose), were developed as a new class of antiviral polyanions [26], but again, their antiviral potential was not thoroughly evaluated. Since the original description of PAA and COAM, polycarboxylates have not received further attention as either IFN inducers or antivirals in general.

**Figure 1 viruses-16-00169-f001:**
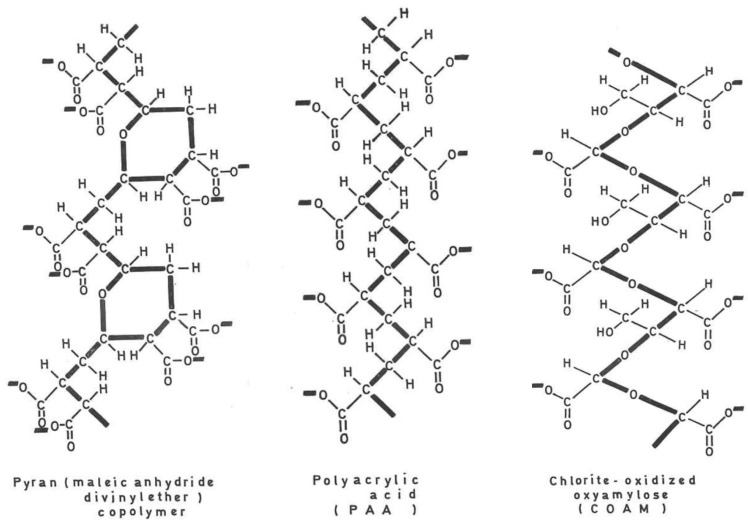
Interferon inducers (pyran copolymer, PAA and COAM).

## 3. Poly(I).poly(C) (Figure 2)

The year 1967 was important for the interferon field, characterized by the crucial observation by Maurice Hilleman and his colleagues at Merck that interferon could be induced by double-stranded RNA from mycophages (*Penicillium stoloniferum*: statolon [27], *Penicillium funiculosum* [28]), synthetic origin [i.e., poly(I).poly(C)] [29], double-stranded RNA from reovirus [30] and double-stranded RNA from MS2 coliphage [31]. That poly(I).poly(C) was such a powerful inducer of interferon drew my attention and tempted us (in vain) to synthesize new analogues of poly(I).poly(C) that were either more potent or less toxic (or both) than the parent compound. However, our attempts to dissociate the interferon-inducing capacity of poly(I).poly(C) from its toxicity by structural modifications invariably failed [32,33].

However, we were successful in employing poly(I).poly(C) to induce the messenger RNA for human fibroblast IFN (IFN-β), which then proved crucial for the cloning of the IFN-β gene [34] and its expression [35]. As an extra bonus, the biotechnology developed in Walter Fiers’ laboratory allowed us to clone and express a by-product of IFN-β, then baptized as IFN-β2 [36] and now better known as IL-6 (interleukin type 6). While poly(I).poly(C) was never licensed for the treatment of any human disease, IFN-β found its principal therapeutic niche in the treatment of multiple sclerosis (MS).

**Figure 2 viruses-16-00169-f002:**
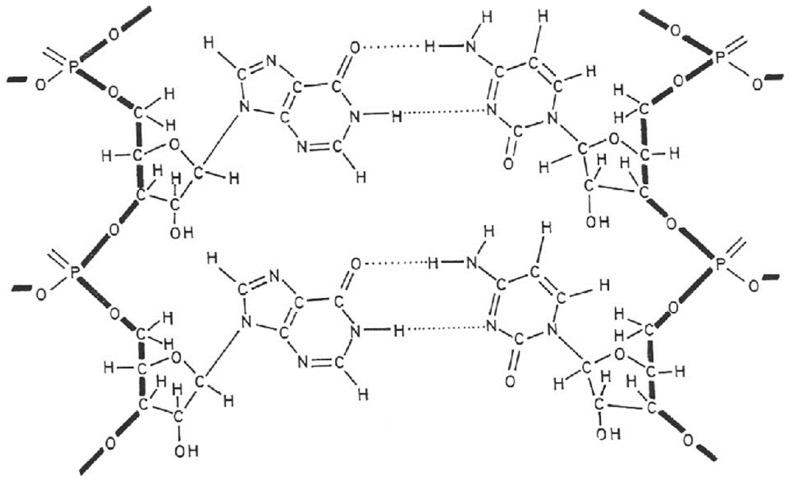
Interferon inducer poly(I).poly(C).

## 4. Suramin (Figure 3)

Suramin was the first drug ever reported to inhibit the replication of the AIDS virus (then still called HTLV-III (LAV)) [37]. Suramin had already been used in the treatment of sleeping sickness (infection caused by Trypanosoma gambiense and transmitted by bites of the tsetse fly (*Glossina palpalis*)] since the early 1920s). That it was evaluated against the AIDS virus stems from the observation I personally performed in the mid-1970s and published in 1979 [16] on the advice of the Journal’s Editor at that time, R.C. Gallo. Suramin was then further evaluated in AIDS patients and found to suppress the replication of HTLV-III (LAV) in vivo [38]. Since AZT had been discovered in the meantime, and suramin had an undesirable toxicity for several organs, suramin was not further pursued in the treatment of AIDS patients. The presumed target of action (the HIV RT) was questioned [39], and virus adsorption was instead postulated to explain the main mode of action of the compound. Mitsuya et al. [40] and Baba et al. [41] almost simultaneously reached this conclusion. It was not unexpected that polyanionic substances in casu polysulfonates such as suramin (a hexasulfonate), akin to other polyanions (i.e., polysulfates), would target the virus adsorption process [42].

**Figure 3 viruses-16-00169-f003:**
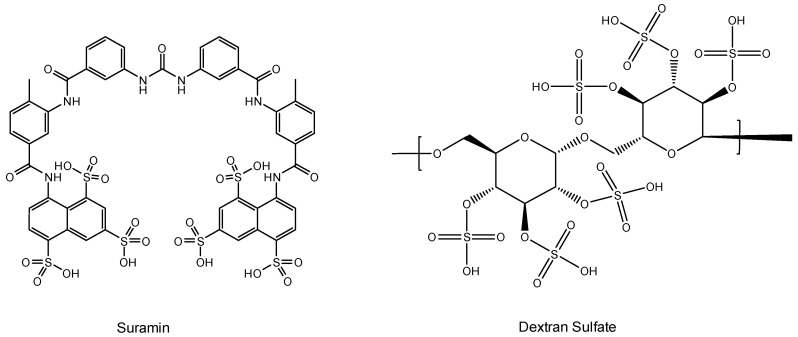
Suramin and dextran sulfate.

## 5. AZT (Figure 4)

Although it was credited as the first anti-HIV agent ever described, AZT (2′,3′-dideoxy-3′-azido thymidine [43], zidovudine, Retrovir) was preceded by suramin [37]. The mechanism of action of AZT, involving phosphorylation to its 5′-triphosphate, which then acted as an inhibitor (chain terminator) of the HIV RT, was resolved by Furman et al. [44], and the compound was formally approved for clinical use for the treatment of AIDS (HIV infections) by the US FDA in 1987, after it had been the subject of a thorough clinical evaluation detailing its toxic side effects [45,46]. As of today, AZT is still available for the treatment of HIV infections, although it has in the meantime been superseded by a vast array of superior (less toxic) antiviral drugs.

AZT was first synthesized by Dr. Jerome Horwitz as a potential anticancer agent [47]. It was evaluated for antiviral activity against HSV, VV and VSV (but found inactive against these viruses) together with a bunch of other nucleoside analogues, sent to me by Fritz Eckstein [48]. HIV, which had not been discovered at that time (the first report on AIDS dates from 1981), was obviously not included in the tests. A murine retrovirus could have been included (as I had experience with both Moloney leukemia virus (MLV) in vitro [49] and Moloney sarcoma virus (MSV) in vivo [50]), but neither MLV nor MSV figured in our panel of test viruses, apparently because of the lack of interest in the medical importance of retroviruses in the late 1970s.

**Figure 4 viruses-16-00169-f004:**
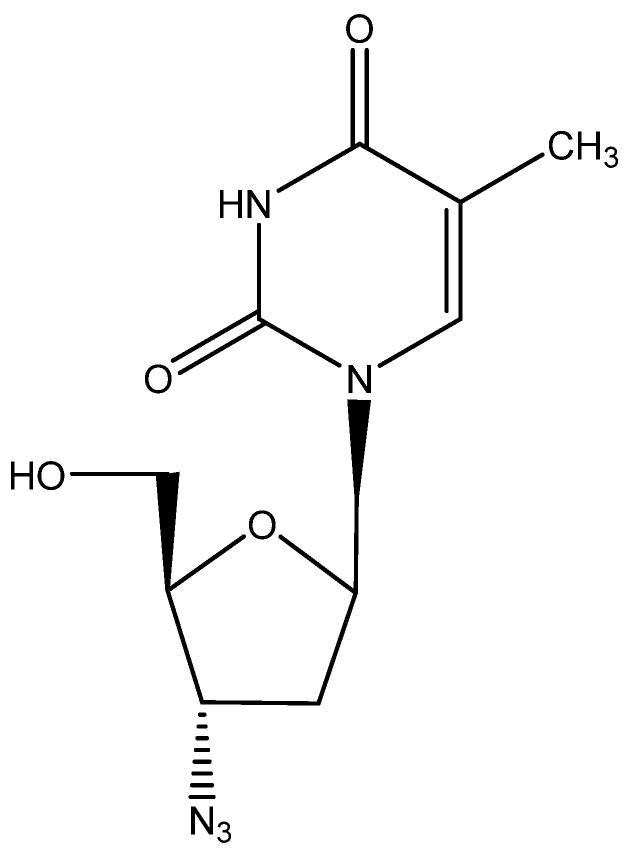
AZT.

## 6. d4T (Figure 5)

Following AZT, various other ddN analogues, i.e., d2I and d2C (which later on would also be approved as anti-HIV drugs), were identified as active inhibitors of HTLV-III/LAV [51]. Also, the publication of Mitsuya et al. in October 1985 incited Piet Herdewijn to embark on the synthesis of a wide variety of ddNs (including d4T, which had initially also been synthesized by Horwitz [52]). These compounds, including d4T (2′,3′-didehydro-2′,3′-dideoxy thymidine), were sent to Jan Balzarini, who was then working in the lab of Sam Broder at NCI; however, using the ATH8 cells as substrate, the anti-HIV activity found was not impressive. It was more so when Masanori Baba repeated the anti-HIV tests in our lab (August 1986). The results were promptly submitted and accepted for publication in Biochemical and Biophysical Research Communications (BBRC) in November 1986. They were finally published in the journal on 15 January 1987 [53]. Later that year, the publications of Hamamoto et al. [54] and Lin et al. [55] followed, confirming that d4T was indeed a powerful inhibitor of HIV replication. The compound, now termed stavudine, would later be commercialized by Bristol Myers as Zerit^®^ and enjoy worldwide acceptance as an anti-HIV drug.

**Figure 5 viruses-16-00169-f005:**
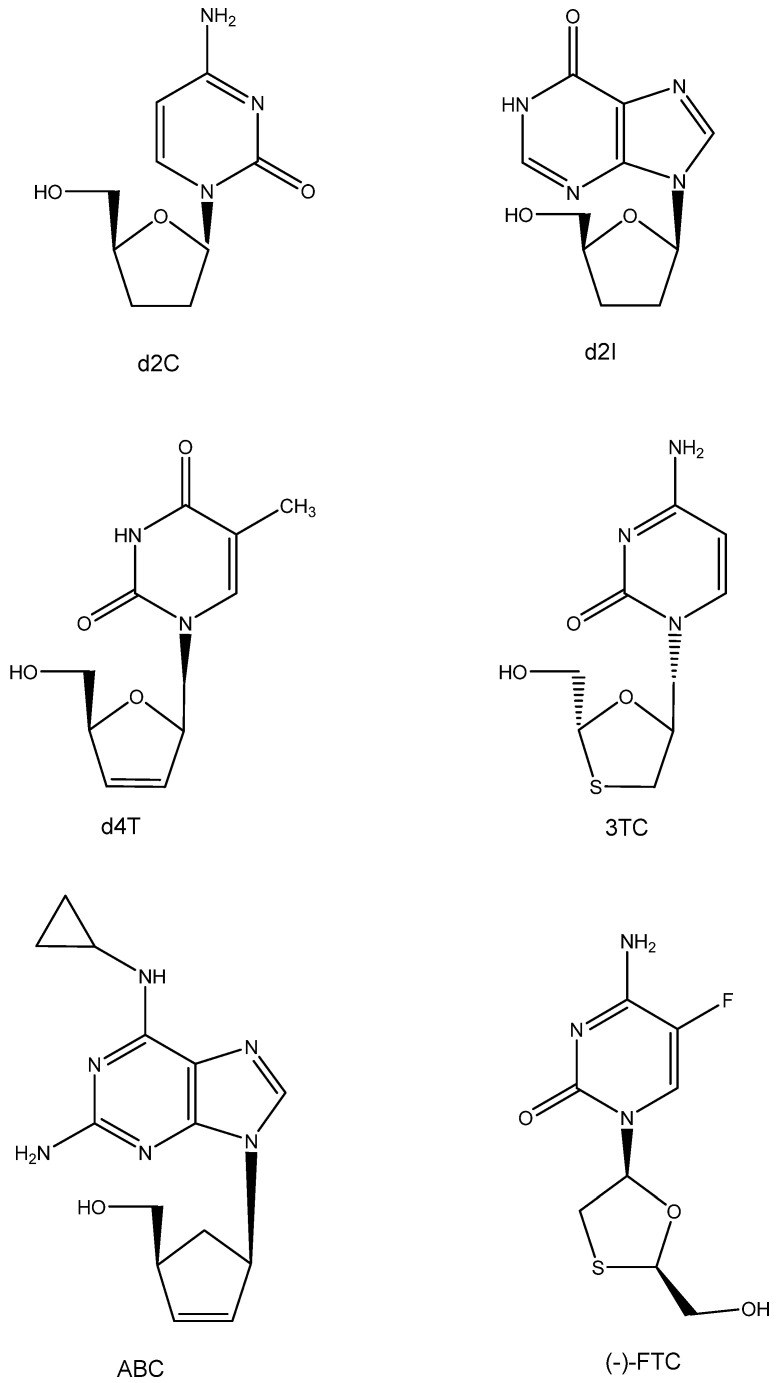
ddNs: d2C, d2I, d4T, 3TC, ABC, and (-)-FTC.

## 7. HEPT and TIBO (Figure 6)

The first compounds ever detected as NNRTIs (non-nucleoside reverse transcriptase inhibitors) were the HEPT [1-[(2-hydroxy-ethoxy) methyl]-6-phenylthiothymine], with TS-II-25 as the leading example. Hiromichi Tanaka from Showa University visited Dick Walker at Birmingham University (UK) in 1986 and delivered a bunch of acyclic pyrimidine analogues, all synthesized at Showa University (Japan). Hiromichi did not dare to send these compounds directly to me, so Richard T. Walker did so on his behalf on a letter head of Showa University, inquiring about my willingness to evaluate these compounds for possible anti-HSV activity. I had already evaluated acyclovir analogues containing a pyrimidine (instead of purine) and had found them inactive as anti-HSV agents. The life of the Showa compounds could have stopped at this stage, but as Masanori Baba had arrived in our lab and had found d4T to be an attractive anti-HIV agent, I passed the compounds to him with the expectation that they would act as negative controls for anti-HIV activity. To my surprise, Masanori Baba found one compound, TS-II-25, active in inhibiting HIV-1 replication (at that time, we did not have access to HIV-2). A second batch proved equally active against HIV-1, and three years later we published these findings in both BBRC [56] and The Journal of Medicinal Chemistry [57], the major problem being that at that time we did not have the slightest clue about the mode of anti-HIV action of these pyrimidine nucleoside analogues, except for the concomitant observation that a collaboration that we had started in 1987 with Janssen Research Foundation had yielded a derivative (R86183, (+)-S-4,5,6,7-tetrahydro-9-chloro-5-methyl-6-(3 -methyl-2- butenyl)- imidazo[4,5,1-jk][1,4]-benzodiazepin2(1H)-thione) (TIBO) [58]. Apparently, the latter compound specifically acted as inhibitor of HIV-1 replication due to a selective inhibitory effect on the HIV-1 RT [58]. The mode of action of the TIBO derivatives was further described in a subsequent article [59]. The HEPT derivatives also acted as specific inhibitors of HIV-1 RT, as ascertained by Baba et al. [60,61]. Despite their different chemical structures, I noticed some resemblance in the conformation of the HEPT and the TIBO derivatives [62], which Dr. Paul Janssen dismissed originally as emanating from my fantasy, but which he must have accepted after Eddy Arnold’s team revealed the butterfly-like structure of both types of compounds [63,64].

What then happened with the clinical developments of both compounds? The HEPT compounds finally led to emivirine (MKC-442) [65], which was originally licensed to Mitsubishi Kasei Corporation (MKC), who transferred it to Wellcome; it proceeded to clinical development as Coactinon^®^ and was finally passed on via Triangle Pharmaceuticals to Gilead Sciences, where its further clinical development ended. The TIBO derivatives followed an equally meandrous path in their development [66]. They finally led to a licensed compound, Rilpivirine (Edurant^®^), that was combined with Tenofovir disoproxil fumarate and emtricitabine (Truvada^®^) or Tenofovir alafenamide and emtricitabine (Descovy^®^) for the treatment and/or prophylaxis of AIDS.

**Figure 6 viruses-16-00169-f006:**
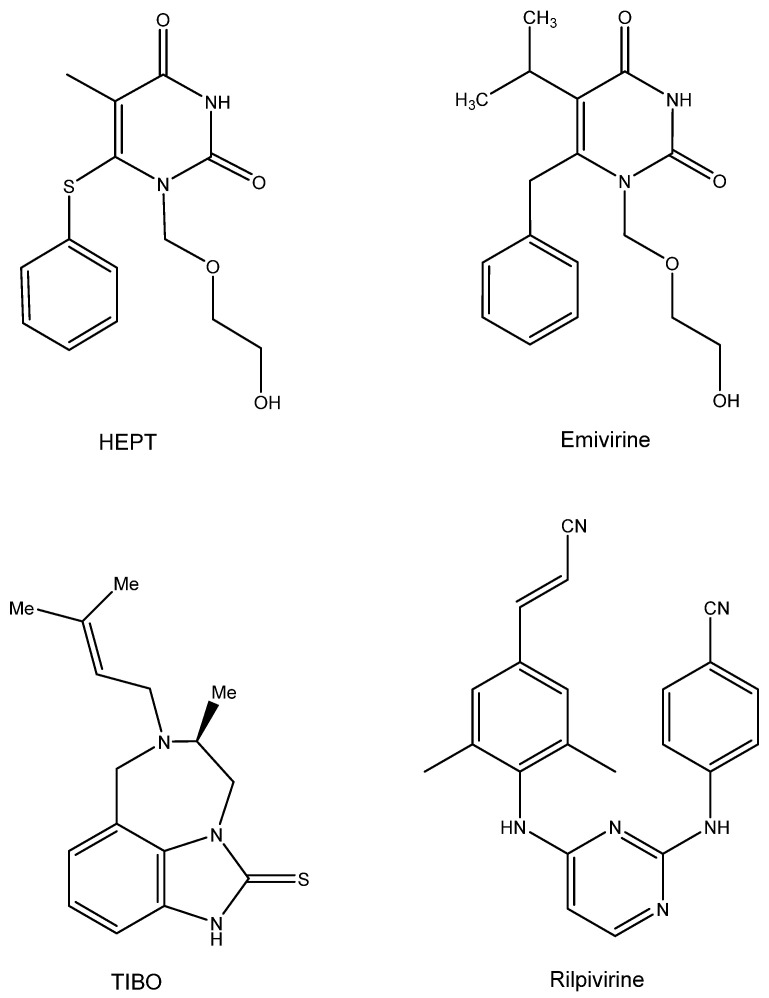
HEPT, Emivirine, TIBO, Rilpivirine.

## 8. DHPA (Figure 7)

In May 1976, more than a year before acyclovir became known for its antiviral properties, i.e., against herpes simplex virus (HSV), I participated in a symposium in Göttingen, Germany (Symposium on Synthetic Nucleosides, Nucleotides and Polynucleotides, Max-Planck-Institut für Biophysikalische Chemie, 3–5 May 1976) where I met Dr. Antonín Holý for the first time and where we agreed to evaluate some compounds that Holý could make available for the study of their antiviral potential. A few months later, I obtained three compounds from Tony, and one of the three compounds, i.e., (*S*)-DHPA (*S*-9-2′,3′-dihydroxypropyladenine), exhibited antiviral activity against several viruses, i.e., vaccinia virus (VV) and vesicular stomatitis virus (VSV). We published our findings in Science [67], a few months after the specific activity of acyclovir against HSV had been published [68], following an earlier announcement of its specific anti-HSV activity in the December 1977 issue of PNAS [69]. (*S*)-DHPA would later be marketed as Duviragel^®^ by the pharmaceutical company Lachema in Czechoslovakia for the topical treatment of fever blisters (HSV labialis). At first, we had no clue as to the mode of action of the compound; it was later ascribed to an inhibitory effect on the S-adenosylhomocysteine hydrolase (SAH hydrolase) and thus interference with the methylation (maturation) of the viral mRNA [70].

The crystal structure of (*S*)-DHPA should have been helpful in deciphering its structural conformation and molecular mode of action, but several attempts to form (*S*)-DHPA crystals invariably failed, until I observed that in an aqueous stock solution kept in the refrigerator for several months, a mold had developed with a needle-like glittering substance in its presence. The university’s crystallographer, Geoffrey King, was informed, and he further examined the purported (*S*)-DHPA crystal. He finally published the results of his examination with Sengier as co-author [71]. Their conclusions can also be found on p. 52 of the first chapter in “Topics in Nucleic Acid Structure—Part 3” [72].

**Figure 7 viruses-16-00169-f007:**
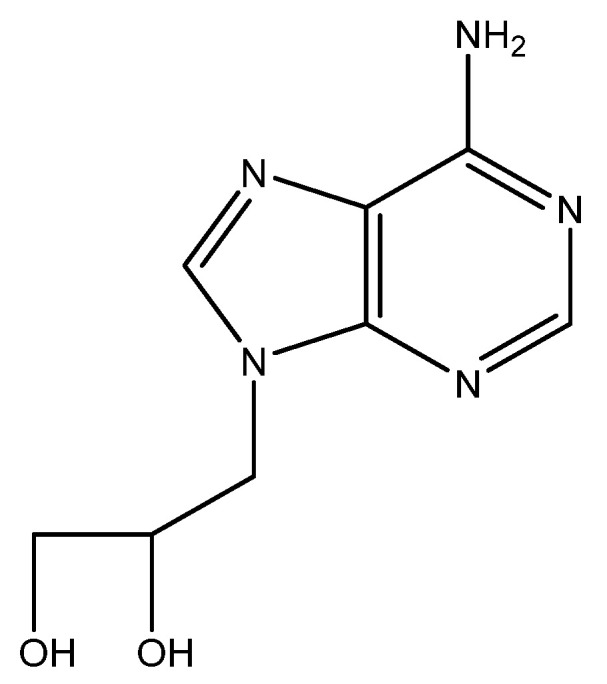
(*S*)-DHPA.

## 9. BVDU (Figure 8)

*E*-5-(2-bromovinyl)-2′-deoxyuridine (BVDU) was first described as an antiviral agent, effective against HSV-1, in 1979 [73]. With the UK branch of Searle, it was agreed to further develop the compound for the treatment of HSV infections. It was then found that BVDU was much less active against HSV-2 than HSV-1 [74], and Searle US decided in 1984 not to further pursue the clinical development of BVDU. Attempts to persuade other pharmaceutical companies to pursue the clinical development of the compound invariably failed, although we had reported the efficacy of the compassionate use of BVDU in the systemic treatment of VZV infections (i.e., herpes zoster) [75] and the topical treatment of HSV eye infections [76,77,78,79,80,81,82,83,84]. Although we were unsuccessful in stimulating any pharmaceutical company into developing BVDU in the Western world, our colleagues in the former DDR were more successful in East Germany and got their BVDU (synthesized locally) marketed by Berlin Chemie for the treatment of herpes zoster in immunosuppressed patients (Helpin^®^). The situation changed in 1989 with the fall of the Berlin wall; BVDU was now made available throughout the whole of Germany (East and West), and Berlin Chemie became part of the Italian company Menarini. Thereupon, BVDU was licensed in several countries all over the world (except for the US and UK) for the treatment of herpes zoster. As a precaution, BVDU should not be administered concomitantly with fluorouracil (FU) or derivatives thereof [85], since the degradation product of BVDU, namely BVU [(*E*)-5-(2-bromovinyl)uracil], was shown to enhance the toxicity of FU, leading to a few cases of death in Japan following the concomitant use of FU and the arabinofuranosyl counterpart of BVDU, BVaraU (sorivudine).

**Figure 8 viruses-16-00169-f008:**
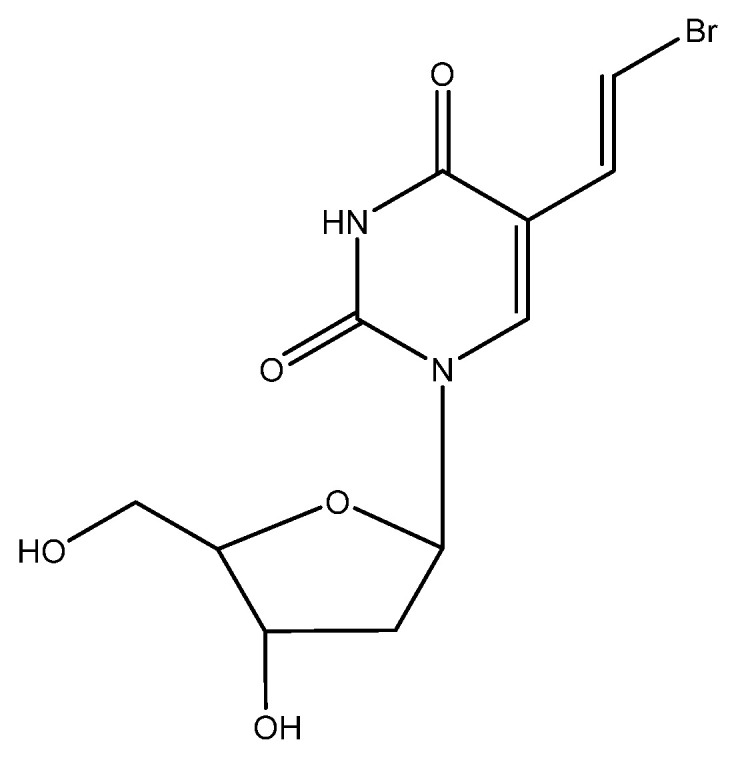
BVDU.

## 10. Aminoacyl Esters of Acyclovir (Figure 9)

The relatively poor solubility of acyclovir in aqueous medium prompted us to design aminoacyl (i.e., alanyl) esters of the compound [86]. The main purpose of this design was to increase the water solubility of acyclovir, so that it would become available as eye drops for the topical treatment of herpetic eye infections (HSV keratitis). This goal was achieved with the glycine ester [87]. Another potential application (that was not accomplished) was to increase the concentration of acyclovir so as to allow its intramuscular or subcutaneous injection. Instead, the valine ester was eventually developed as a substitute for acyclovir itself in the oral treatment of HSV (and VZV) infections [Valtrex^®^ (USA), Zelitrex^®^ (EU)].

**Figure 9 viruses-16-00169-f009:**
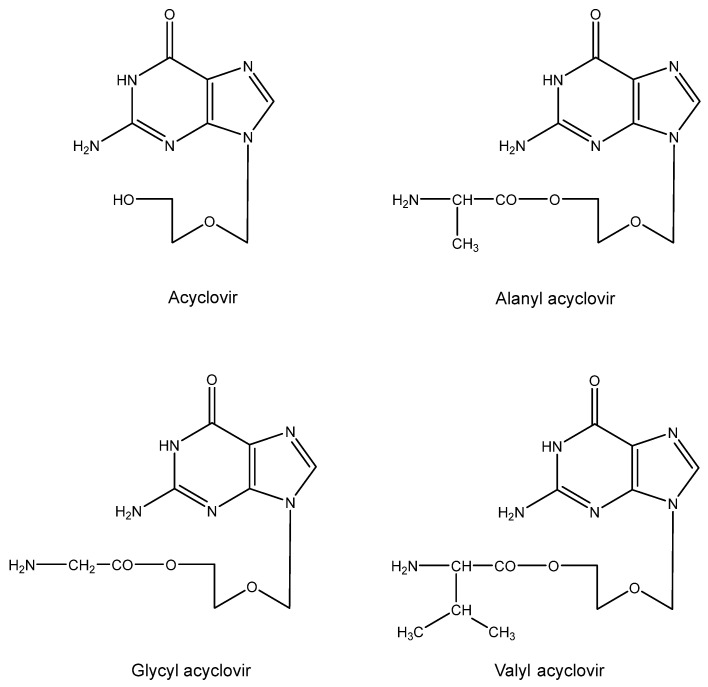
Aminoacyl esters of acyclovir.

## 11. AMD-3100 (Figure 10)

The bicyclam derivation AMD-3100 (initially termed JM3100) belongs to a new class of compounds first identified in 1992 as being active against HIV [88], provisionally presumed to interact with the virus uncoating process. A marked increase in anti-HIV potency was noted when the aliphatic bridge tethering the cyclam rings was replaced by an aromatic (i.e., bis methylphenyl) bridge [89]. The HIV gp120 was then identified as the target for the anti-HIV action of AMD-3100 [90]. It appeared to be an indirect target, the direct target being the CXCR4 co-receptor for HIV entry into the host cells [91,92,93,94].

Phase 1 clinical trials carried out with AMD-3100 revealed an unusual side effect, namely that it caused an increase in the white blood cell (WBC) counts [95]. On close inspection, the elevated WBC counts appeared to consist primarily of hematopoietic stem cells. Thus, AMD-3100 (in the meantime called plerixafor) was recognized as a hematopoietic stem cell mobilizer, and since 2009 it has been marketed as Mozobil^®^ in the autologous transplantation of hematopoietic bone marrow cells in patients with NHL (Non-Hodgkin’s Lymphoma) or MM (Multiple Myeloma) [96,97,98,99].

**Figure 10 viruses-16-00169-f010:**
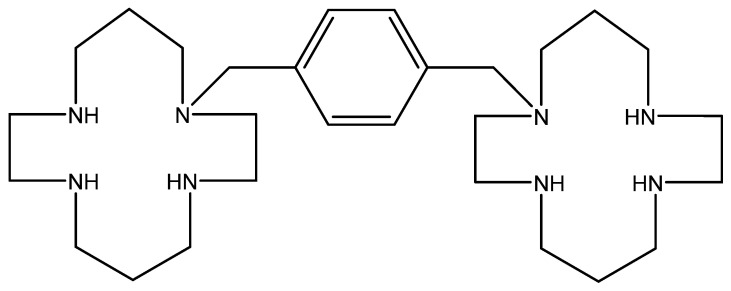
AMD-3100.

## 12. ANPs (Acyclic Nucleoside Phosphonates) (Figure 11)

The era of the ANPs started with the discovery of the broad-spectrum anti-DNA virus activity of (*S*)-HPMPA [9-(3-Hydroxy-2-phosphonylmethoxypropyl)adenine] and antiretroviral activity of PMEA [9-(2-phosphonylmethoxyethyl)adenine] [100]. Then, in 1987, the description of the antiviral activity of several other ANPs followed, including (*S*)-HPMPC (cidofovir) [101], which would be approved ten years later (and marketed as Vistide^®^) for the treatment of human cytomegalovirus (HCMV) retinitis in AIDS patients. As an antiretroviral agent, PMEA (adefovir) was first pursued for the treatment of HIV infections, but as it proved more potent (at lower doses) against hepatitis B virus (HBV) infections, it was eventually marketed (in 2002) as its prodrug, adefovir dipivoxil (Hepsera^®^), for the treatment of chronic hepatitis B. The anti-HIV activity of (*R*)-PMPA [(R)-9-(2-phosphonomethoxypropyl)adenine] was first described in 1993 by Balzarini et al. [102]. That it would be superior to AZT in the treatment of retrovirus infections became evident from the results of Tsai et al. in SIV (simian immunodeficiency virus) infections in rhesus macaque monkeys [103]. The results of Tsai et al. could be interpreted as a prelude of the prophylactic effect of (*R*)-PMPA (now dubbed tenofovir) on HIV infection, for which the compound, in combination with emtricitabine, finally received US FDA approval on 16 July 2012, the exact date on which the co-inventor of the ANPs, Antonín Holý, died. Meanwhile, tenofovir had been derivatized to its oral prodrug, tenofovir disoproxil [104,105], and upon the further addition of fumarate, tenofovir disoproxil fumarate (TDF), it received final approval by the USA FDA in 2001 for the treatment of HIV infections. This was followed in 2004 by the approval of TDF in combination with emtricitabine (marketed as Truvada^®^) and in further combination with efavirenz in 2006 (marketed as Atripla^®^). Other combinations containing TDF have been reviewed [106].

Meanwhile, Lee et al. [107] described a new prodrug of tenofovir, TAF (tenofovir alafenamide), that has also been the subject of various combinations reminiscent of those applied to TDF, and these combinations have also been reviewed previously [106]. Akin to the combination of TDF with emtricitabine (marketed as Truvada^®^), the combination of TAF with emtricitabine has been marketed as Descovy^®^ for the prophylaxis of HIV infections.

**Figure 11 viruses-16-00169-f011:**
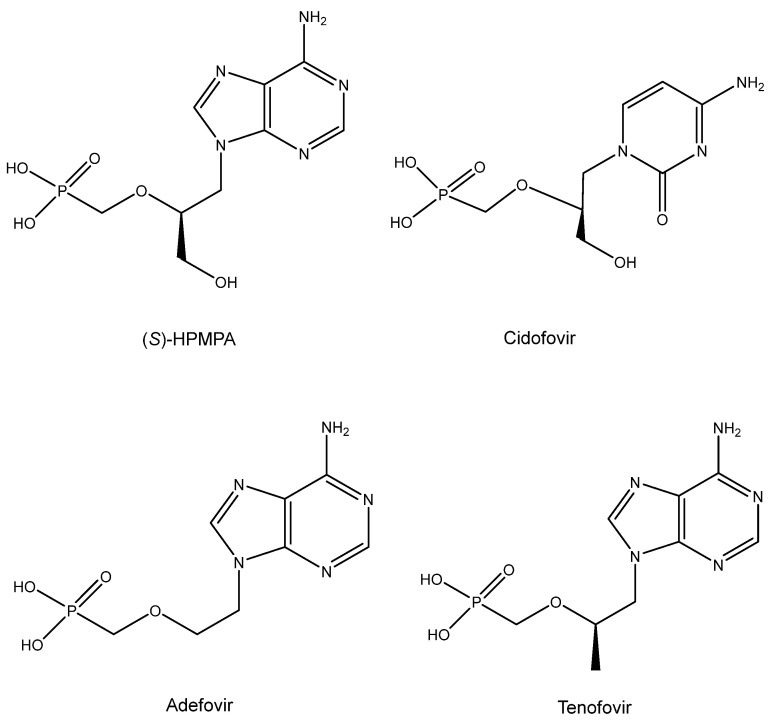
ANPs: (*S*)-HPMPA, Cidofovir, Adefovir and Tenofovir.

## 13. Conclusions

The main conclusion in antiviral drug development is that the intended goals were often not achieved. A prominent exception to this rule is acyclovir, which was originally discovered to be an anti-HSV agent and has been finally used and marketed for the treatment of HSV-1 and HSV-2 infections, as has been the valine ester of acyclovir, valacyclovir. Interferon inducers, such as polyacrylic acid (PAA) and even poly(I).poly(C), did not attain any medical use, and interferon-β, which poly(I).poly(C) helped to clone, got mainly applied in the treatment of multiple sclerosis (MS). While originally discovered as an inhibitor of HSV-1, BVDU obtained its therapeutic application mainly in the (oral) treatment of VZV infections, i.e., herpes zoster. Adefovir (PMEA) was first pursued for its anti-HIV activity before it was eventually commercialized as its (oral) prodrug adefovir dipivoxil (Hepsera^®^) for the treatment of HBV infections. The oral prodrugs of tenofovir TDF and TAF fulfilled their initial goals in that they have proved to be therapeutically and prophylactically useful (the latter when combined with emtricitabine) against HIV, but, additionally, they have also proven to be amenable for the treatment of HBV infections. AMD-3100 represents a case where antiviral drug development efforts have led to a spin-off in a totally different field, since it shifted from an anti-HIV agent to a stem cell mobilizer (Mozobil^®^) in the therapy of cancer.

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
