# Peer review of "Selected Milestones in Antiviral Drug Development"

_viruses, 2024, doi:10.3390/v16020169_

Round 1

Reviewer 1 Report

Comments and Suggestions for Authors

The author summarized the development of antiviral drugs, including major milestones and the contributions from his laboratory. This review provides a brief history of antiviral drug development, and more importantly insight about the oftentimes mismatch between intended indications at the beginning of discovery/development and actual applications of drug molecules by the end. It also pointed out that seemly reasonable assumptions may turn out to be false.

I recommend the manuscript be accepted for publication in present form.

Author Response

I am most grateful for this Reviewer's positive comments.

Reviewer 2 Report

Comments and Suggestions for Authors

This is a very nice reading with some historical point of view of some of the author's contribution to the discovery of new drugs. Given the importance of Dr. De Clercq to the field, this is a must read for the generation of young scientists/virologists. My main recommendation would however to modify the title, because it is a little bit misleading. Indeed, “Milestones in antiviral drug development", would suggest a broad review that deals with all milestones such as approval sand discovery of the first protease inhibitors which completely changed the clinical management of HIV-1 infected patients first, and HCV infected patients later on. Similarly, discovery of integrase inhibitors would need to be mentioned (Bictegravir, is part of Biktarvy, the most blockbuster antiviral drug worldwide, and integrase inhibitors are commonly part of most popular anti HIV-1 regimen). At the same time other drugs which clearly made an impact on the management of HCV should be discussed (Sofosbuvir and NS5A inhibitors such as daclatasvir). An so on so forth. Indeed, changing title would resolve all issues.

Author Response

  • I agree with Reviewer 2 that the title should be adapted. I suggest to change it to "Selected milestones in antiviral drug development".
  • As the Reviewer observed, the HIV protease inhibitors have not been discussed; the HIV integrase inhibitors (such as bictegravir) have already been dealt with in another article (De Clercq E, Zhang Z, Huang J, Zhang M, Li G. Biktarvy for the treatment of HIV infection: Progress and prospects. Biochem Pharmacol. 2023 Nov;217:115862. doi: 10.1016/j.bcp.2023.115862. Epub 2023 Oct 17. PMID: 37858869).
  • For the HCV inhibitors such as sofosbuvir, this has also been described previously (Li G, De Clercq E. Current therapy for chronic hepatitis C: The role of direct-acting antivirals. Antiviral Res. 2017 Jun;142:83-122. doi: 10.1016/j.antiviral.2017.02.014. Epub 2017 Feb 24. PMID: 28238877; PMCID: PMC7172984).

Reviewer 3 Report

Comments and Suggestions for Authors

This review article describes various inhibitors the author developed as therapy of viral infections. The article describes the history and development of multiple drugs and gives valuable information to the readers. However, there are extensive language corrections and editing required for publication of this review.

It’s better to write a review in the view of scientific merits rather than the grounds of personal experiences. I strongly suggest avoiding line 14-15, 54-55, 73. Please avoid personal history lines, 116-117, 135-136, 185 and many such sentences that do not add value to the science and are only good for bio sketch. 

Line 195 should be HSV?

Comments on the Quality of English Language

There are extensive language corrections and editing required for publication of this review. Hard to list each lines

Author Response

I strongly considered to delete lines 14 and 15 and others suggested by Reviewer 3, but I noticed that leaving out 14 & 15 would strongly handicap the introduction and then I decided not to do so.

I agree that there are some lines that do not necessarily add value to the science and are only good for bio sketch, but what is wrong with a bio sketch?

In line 195, "HVSV-1" should indeed be corrected to "HSV-1".

That Reviewer 3 is questioning my English command is rather surprising, as the other Reviewers thought my English is good. Also, Reviewer 3 failed to indicate any specific language corrections.

Reviewer 4 Report

Comments and Suggestions for Authors

This extensive, entertaining and historical review is written by one of the giants in the field who contributed remarkably into the study of viruses and development of new antivirals. It was a pleasure and a good lesson to read this paper and get to know the history of antiviral drug development from one of the major historical figures. The main message that clearly shines throughout the review is that the original goals are rarely achieved, but helps to develop new, unexpected medications for different human diseases.

I am happy to recommend the review for publication.

There is a few minor issues that would be helpful to correct:
Figures throughout the manuscript - are they original or need to be referenced?

Line 76: AZT abbreviation should be given at first mention

Dr. De Clercq mentions HLTV-III as the original name for the HIV, but does not provide information when it was baptised as "HIV"

Line 109: following - from the capital letter

111: HLTV-III/LAV - abbreviation is not given.

In general, there is a mix of HTLV-III and HIV. I would suggest to make the use of these terms more consistent.

Author Response

I would like to thank Reviewer 4 very much for his comments.

Minor issues:

  1. All the (original) figures concern chemical structures and they do not need to be referenced.
  2. Line 76: the AZT abbreviation (Fig. 4) was given with the further description in analogy with the other ddNs (Fig. 5).
  3. The name HIV dates from 1987; before that date it was named HTLV-III/LAV. HTLV-III is "human T cell leukemia type III" and LAV means "lymphadenopathy associated virus".
  4. In line 109, the word "following" should indeed be capitalized.